# Deciphering the Microbiological Mechanisms Underlying the Impact of Different Storage Conditions on Rice Grain Quality

**DOI:** 10.3390/foods13020266

**Published:** 2024-01-15

**Authors:** Zhuzhu Qiu, Fenghua Wu, Hao Hu, Jian Guo, Changling Wu, Peng Wang, Jiangang Ling, Yan Cui, Jing Ye, Guanyu Fang, Xingquan Liu

**Affiliations:** 1College of Forestry and Biotechnology, Zhejiang A&F University, Hangzhou 311300, China; 2021102032008@stu.zafu.edu.cn; 2College of Food & Health, Zhejiang A&F University, Hangzhou 311300, China; wufh@zafu.edu.cn (F.W.); 20180015@zafu.edu.cn (H.H.); jguo@zafu.edu.cn (J.G.); wuchangling@zafu.edu.cn (C.W.); wpeng@zafu.edu.cn (P.W.); 3National Grain Industry (High-Quality Rice Storage in Temperate and Humid Region) Technology Innovation Center, Zhejiang A&F University, Hangzhou 311300, China; 4Ningbo Academy of Agricultural Sciences, Ningbo 315000, China; nbnjg@163.com (J.L.); cuiyan1605@126.com (Y.C.); 5Zhejiang Tongqu Grain Storage Co., Ltd., Quzhou 324000, China; yejing20024060637@163.com

**Keywords:** rice grains storage, mycotoxins, microbial communities

## Abstract

Different storage conditions can influence microbial community structure and metabolic functions, affecting rice grains’ quality. However, the microbiological mechanisms by which different storage conditions affect the quality of rice grains are not yet well understood. This study monitored the quality (the content of starch, protein, etc.) and microbial community structure of rice grains stored under different storage conditions with nitrogen gas atmosphere (RA: normal temperature, horizontal ventilation, RB: normal temperature, vertical ventilation, RC: quasi-low temperature, horizontal ventilation). The results revealed that the rice grains stored under condition RB exhibited significantly lower quality compared to condition RA and RC. In addition, under this condition, the highest relative abundance of *Aspergillus* (16.0%) and *Penicillium* (0.4%) and the highest levels of aflatoxin A (3.77 ± 0.07 μg/kg) and ochratoxin B1 (3.19 ± 0.05 μg/kg) were detected, which suggested a higher risk of fungal toxin contamination. Finally, co-occurrence network analysis was performed, and the results revealed that butyl 1,2-benzenedicarboxylate was negatively correlated (*p* < 0.05) with *Moesziomyces* and *Alternaria*. These findings will contribute to the knowledge base of rice storage management and guide the development of effective control measures against undesirable microbial activities.

## 1. Introduction

Rice, as one of the most important staple crops globally, plays a significant role in human nutrition and food security [1]. However, during the storage process, rice is often plagued by microbial contamination, leading to a decline in quality and potential food safety issues [2]. Understanding the impact of different storage conditions on microbial community dynamics and subsequent effects on rice quality is crucial for ensuring food security and optimizing storage practices. Microorganisms, including bacteria, fungi, and yeasts, are ubiquitous in the environment and can readily colonize rice grains [3]. The composition and activity of microorganisms in rice can be influenced by various factors, such as temperature, humidity, storage duration, and initial microbial loads [4,5]. These factors collectively determine the growth, survival, and metabolic capabilities of microorganisms, which ultimately shape the microbial community structure in stored rice.

The microbial community in rice grains is highly complex, comprising both beneficial and detrimental microorganisms. Detrimental microorganisms, including pathogenic bacteria, molds, and mycotoxin-producing fungi, can cause spoilage, quality deterioration, and health risks [6,7]. Microorganisms can use the nutrients of rice, which will affect the integrity of the rice, resulting in reduced germination rate. Meanwhile, rice will produce an unpleasant smell after being decomposed by microorganisms and can be utilized by mold to produce a moldy taste. The appearance, color, and shape of rice are affected by microbial community activities. Research has shown that different storage conditions significantly impact the microbial community dynamics in rice [8]. Related research found that a rice grain storage warehouse adopted different storage methods such as ventilation, nitrogen gas atmosphere, etc., to ensure the good quality of rice grains and inhibit the growth of mold. Temperature and humidity play pivotal roles in influencing microbial growth and survival [9]. Higher temperature and moisture levels can create a favorable environment for microorganisms, leading to increased microbial activity and potential spoilage [10]. Previous studies have demonstrated that temperature and relative humidity influence microbial growth [11,12]. Conversely, low temperatures and dry conditions can inhibit microbial growth but may not completely eliminate microorganisms [13,14]. Understanding how varying storage conditions affect the balance between beneficial and detrimental microorganisms is crucial for maintaining rice quality and safety. Furthermore, the presence of specific microorganisms in rice can have profound implications for its quality attributes [15]. Microorganisms interact with rice grains through various mechanisms, including enzymatic activities and metabolic processes. These interactions can lead to changes in texture, flavor, aroma, and nutritional composition. For instance, certain microorganisms can modify complex carbohydrates, proteins, and lipids, impacting the overall nutritional profile and digestibility of rice. Moreover, the production of metabolites and enzymes from microorganisms can contribute to off-flavors, off-odors, and the development of mycotoxins, compromising rice quality and consumer acceptability [16]. Various molds, including *Aspergillus*, *Penicillium*, and others, can cling to harvested rice grains through exposure to wind and rain. For example, cereal infected by *Fusarium* spp. and stored under simulated conditions for six weeks exhibited excessive levels of Deoxynivalenol and Zearalenone, which compromised the degradation of starch and protein during storage [17,18].

In light of the intricate relationship between storage conditions, microbial dynamics, and rice quality, it is crucial to comprehensively investigate these conditions to develop appropriate storage strategies that preserve rice quality and ensure food safety. This research aims to provide valuable insights into the microbial community dynamics in stored rice under different storage conditions and shed light on their effects on rice quality attributes. These findings will contribute to the knowledge base of rice storage management and guide the development of effective control measures against undesirable microbial activities.

## 2. Materials and Methods

### 2.1. Collection of Samples

Rice grain samples were collected from four grain depots from Ningbo, Quzhou, Shaoxing, and Hangzhou cities in Zhejiang Province, China. Hangzhou is located at 29°11′–30°33′ N and 118°21′–120°30′ E. The four grain depots are close. All samples were japonica rice and stored for about one year. The four granary types are bungalows, and the grain stack height of the stored rice is about 4~5 m high. Four granaries have taken nitrogen gas control measures to control the growth of insects and microorganisms. The grain depots in Quzhou and Shaoxing are stored at room temperature, and the average temperature of the grain pile is not more than 25 °C throughout the year. The grain depots in Hangzhou and Ningbo adopt quasi-low temperature storage, the temperature is controlled by air conditioning, and the average temperature of grain stacks does not exceed 20 °C throughout the year. During the storage of rice, ventilation will be used to ensure the stability of the temperature and humidity of the entire grain depot. Quzhou grain depot adopts the way of vertical ventilation, which means through the exhaust pipe and the inlet pipe connected up and down, to achieve the circulation of hot and cold airflow. Horizontal ventilation is adopted in Hangzhou, Ningbo, and Shaoxing, which means the ventilation duct is placed horizontally on the ground, so that the airflow is inhaled from one side of the tuyere and discharged from the other side of the tuyere after passing through the grain pile horizontally. A total of 13 rice grain samples collected from 4 different grain depots were surveyed to estimate differences in various storage conditions (RA: normal temperature, horizontal ventilation, nitrogen gas atmosphere; RB: normal temperature, vertical ventilation, nitrogen gas atmosphere; RC: quasi-low temperature, horizontal ventilation, nitrogen gas atmosphere). The average temperature and humidity of four grain depots on one day in winter were detected and are shown in Table 1.

### 2.2. DNA Extraction, Sequencing, and Bioinformatics Analysis

For sample preparation, rice grains were collected via three-layer and five-point sampling methods. Then, 5 g of each sample with three replicates per treatment was transferred to EP tubes. The microorganisms on the rice grain surface were extracted with sterilized water under sterile operating conditions. The V3–V4 region of the bacterial 16S rRNA gene was amplified using primer sets 338F (5′-ACTCCTACGGGAGGCAGC A-3′)/806R (5′-GGACTACHVGG GTWTCTAAT-3′). In addition, the ITS1 region (internal transcribed spacer) of the fungal communities was amplified using primer sets ITS1F (5′-CTTGGTCATTTA-GAGGAAGTAA-3′)/ITS2R (5′-GCTGCGTTCT TCATCGATGC-3′) [19]. The amplified products were detected on an Illumina NovaSeq platform (Novogene Co., Ltd., Beijing, China).

QIIME2 was used to qualify the obtained raw sequences [20]. The chimera sequences were removed from the sequences, which were compared with the Sliva database using the UCHIME algorithm [21,22]. The clean data were denoised into amplicon sequencing variants (ASVs) via QIIME2. The Sliva database was used to annotate for the representative sequences. R 3.6.0 was used to perform diversity and statistical analyses [23].

### 2.3. Determination of Physical and Chemical Indicators

#### 2.3.1. Detection of Ochratoxin A and Aflatoxin B1 in Rice Grain Samples

The samples were washed with 10 mL of methanol aqueous solution (7:3, *v*/*v*). The concentration of ochratoxin A (OTA) and aflatoxin B1 (AFB1) was detected by using an ELISA kit (Suwei Microbiology Research, Wuxi, China), according to the manufacturer protocol [24].

#### 2.3.2. Moisture Content Detection

According to the method [25], the moisture content of the samples was measured according to the manufacturer protocol. A total of 3.0 g of rice grain was weighed and put into an aluminum box. Then, the sample was put in an oven to bake at 105 °C for 3 h. After baking, the sample was taken and cooled with a lid, baking again. We took out the sample after cooling, weighed it every 30 min, and baked it until the difference between the two weights was not more than 0.002 g.

#### 2.3.3. Starch Content Detection

The amount of starch content was determined via the enzymatic hydrolysis method [26]. The milled rice grain flour was transferred to a beaker after removing fat and soluble sugar using petroleum ether and 85% ethanol (Sinopharm Chemical Reagent Co., Ltd., Shanghai, China), respectively, and the filter paper was washed with 50 mL of water. After heating for 15 min, 20 mL enzyme (Sinopharm Chemical Reagent Co., Ltd., Shanghai, China) solution was added for enzymatic hydrolysis. We took 50 mL filtrate and added 5 mL hydrochloric acid to reflux for 1 h. The sample was determined via alkaline copper tartrate solution. 

#### 2.3.4. The Content of Soluble Proteins Detection

According to the method of Fang et al. [27,28], 5.0 g of milled rice grain was put into 20 mL of petroleum ether (Shanghai McLean Biochemical Technology Co., Ltd., Shanghai, China). After drying, 1.0 g of the defatted rice grain was taken, extracted with pure water at 45 °C, and then centrifuged at 10,000 r/min for 10 min to obtain the albumin. After water extraction, the flour was extracted with 5% NaCl (Shanghai McLean Biochemical Technology Co., Ltd., Shanghai, China) at 40 °C to obtain the globulin. Then, the remaining flour was extracted with 70% ethanol at 45 °C, and the sediment was extracted with 1 mL of 0.2% NaOH at 50 °C to obtain gliadin and glutenin. Each treatment needed three replicates. The proteins were detected via the Braford method. 

#### 2.3.5. Fatty Acid Value Detection

The fatty acid value of rice grain samples was detected via an BLH-840K automatic fatty acid value analyzer (Zhejiang Bethlehem Apparatus company, Ltd., Taizhou, China).

#### 2.3.6. Volatile Component Detection

The volatile components were analyzed using headspace–solid-phase microextraction (HS-SPME) coupled with gas chromatography–mass spectroscopy (GC-MS) (Shimadzu Co., Kyoto, Japan). DVB/CAR/PDMS-coated SPME fibers (50/30 μm) were chosen to extract volatile components with helium, which was used as a carrier gas at a flow rate of 1.0 mL/min. The procedure was divided into four stages. The initial temperature was 50 °C and was maintained for two minutes, and then it increased to 125 °C and lasted for 3 min at a rate of 8 °C/min. Continuously, the temperature rose to 165 °C at a rate of 4 °C/min and was kept for 2 min, and then it finally increased to 230 °C/min at a rate of 10 °C/min and was maintained for 2 min. The MS conditions were electron energy 70 eV, electron impact (EI) ion source temperature 280 °C, and transmission line temperature 230 °C. According to the method in [29], 2.0 g of rice grain was weighed and placed into a 20 mL headspace glass sampling vial. Before the measurement, 20 μL of internal standard Decanoic acid ethyl ester (8.63 μg/mL, Aladdin, China) was added to each vial. Then, the sample was preheated for 1 h at 40 °C.

### 2.4. Statistical Analysis

The statistical analysis was performed using the Origin Pro 8.6 program (SAS Inst. Inc., Gary, NC, USA) and analyzed via one-way analysis of variance (ANOVA) with a post hoc Duncan test using SPSS 22.0. Duncan’s multiple range test (*p* < 0.05) was used for the analysis of significant difference. Volatile components with VIP values greater than 1 in the PLS-DA model were constructed to visualize using a free online website (http://www.ehbio.com/test/venn/#/, accessed on 5 September 2023). All experiments were replicated at least twice, and the data are expressed as means ± standard deviations (SD).

## 3. Results and Discussion

### 3.1. Determination of Physicochemical Indices

Under various storage conditions, rice grains can exhibit differences in their biochemical components. The current study aimed to compare the quality of rice grains stored under different conditions. Moisture content serves as a primary indicator for assessing the quality of rice grains, which is closely linked to storage practices. Excessive moisture in a paddy can potentially elevate the risk of mold formation and simultaneously undermine its germination capability. To optimize storage conditions and ensure the overall quality of rice grains, it is crucial to maintain an appropriate moisture level [30]. The moisture content of rice grains stored under normal temperature, vertical ventilation, and nitrogen gas atmosphere was 11.11%, which was the lowest compared with other storage conditions (Figure 1A). Additionally, the starch content and glutenin levels were also the lowest, at 46.93 g/100 g and 2.99 mg/g (Figure 1C,G), respectively. Moreover, the protein levels, including albumin, globulin, gliadin, and glutenin, were found to be the lowest in rice grain stored under normal temperature, vertical ventilation, and nitrogen gas atmosphere conditions (albumin: 0.47 mg/g, globulin: 1.15 mg/g, gliadin: 0.40 mg/g, glutenin: 2.99 mg/g) when compared to the other two storage conditions. Previous studies have indicated that storing rice at low temperatures and low humidity can effectively reduce enzyme activity. These studies have also shown that after 100 days of storage under such conditions, the quality of rice remains largely unchanged and comparable to the freshly harvested sample [31,32]. This suggests that maintaining low temperature and humidity levels during storage can help preserve the quality of rice for an extended period. The respiration and enzyme activity of rice can cause changes in its quality. High temperature and humidity can accelerate these processes, leading to a deterioration in rice quality. It is important to control the temperature and humidity at lower levels during storage to minimize these effects and maintain the quality of the rice [33]. Furthermore, rice grains stored under normal temperature, horizontal ventilation, and nitrogen gas atmosphere conditions exhibited the highest starch content (56.42 g/100 g) and glutenin levels (4.06 mg/g) among the three storage environments. It is worth noting that the fatty acid value is an important indicator used to assess the freshness of rice. It is widely accepted that a higher fatty acid value indicates a more pronounced deterioration in paddy quality [34]. In the case of rice grains stored under normal temperature, horizontal ventilation, and nitrogen gas atmosphere, the fatty acid value was measured at 11.87 mg/100 g, which represents the smallest amount of deterioration in rice grain quality (Figure 1B). During rice grain storage, the variation in fatty acids was mainly caused in two ways, including oxidation and hydrolysis. During hydrolysis, the change in fatty acid value is mainly due to the combined action of the enzyme activity of the rice grain itself and molds [35]. Among the three storage conditions, the fatty acid value of rice grains stored under the conditions of normal temperature, vertical ventilation, and nitrogen gas atmosphere (19.48 mg/100 g KOH) and quasi-low temperature, horizontal ventilation, and nitrogen gas atmosphere (21.46 mg/100 g KOH) reached a higher level. It was considered that these two storage conditions were suitable for microbial growth and adverse for rice grain storage. However, in the study, the fatty acid values of rice grains stored under the three conditions did not exceed the national storage standards for suitability. This indicates that all three storage conditions provide a better assurance of maintaining the quality of rice.

### 3.2. Comparison of Volatile Components of Rice Grains Stored under Different Conditions 

To better investigate the variances among rice grains stored under the three storage conditions, we conducted an analysis specifically targeting the volatile components. These volatile components are crucial contributors to the aromatic characteristics of rice grains [36]. The degradation and mold growth in rice grains can result in the process of unpleasant odors. One of the significant volatile substances produced by fungi in this process is 1-octen-3-ol, along with 3-octanone. These components contribute to the unpleasant odor associated with fungal contamination in rice grains [37,38]. It is worth noting that no mold-related components were detected in rice grains stored under the three storage conditions. The volatile components detected in the rice grains were mainly divided into seven categories (Figure 2A). 

The results showed that the largest composition of volatile components in the three storage scenarios was hydrocarbons whose distribution rate was more than 50%. A total of 39 volatile components were identified in rice grains stored under the condition of normal temperature, horizontal ventilation, and nitrogen gas atmosphere, including 1 alcohol, 8 esters, 1 ketone, 1 aldehyde, 3 alkenes, 23 hydrocarbons, and 2 aromatic and heterocyclic compounds (Table 2). A total of 49 volatile components were detected in rice grains stored under normal temperature, vertical ventilation, and nitrogen gas atmosphere, including 3 alcohols, 9 esters, 1 ketone, 1 aldehyde, 2 alkenes, 30 hydrocarbons, and 3 aromatic and heterocyclic compounds (Table 2). A total of 45 volatile components were detected in rice grains stored under quasi-low temperature, horizontal ventilation, and nitrogen gas atmosphere, including 3 alcohols, 9 esters, 1 ketone, 1 aldehyde, 1 alkene, 30 hydrocarbons, and 0 aromatic and heterocyclic compounds (Table 2). Among these volatile components, fatty acids in rice grains are oxidized to peroxides, mainly generating aldehydes and ketones [39]. The rice grains stored under the conditions of normal temperature, vertical ventilation and quasi-low temperature, horizontal ventilation contain higher levels of ketones (RB: 6.17 μg/kg, RC: 8.67 μg/kg) and aldehydes (RB: 108.45 μg/kg, RC: 61.27 μg/kg). The results also showed the reasons for the higher fatty acid value of rice grains stored under these two storage conditions.

Partial Least Squares Discriminant Analysis (PLS-DA) is a supervised discriminant analysis method that establishes a relationship model between metabolite expression and sample categories through Partial Least Squares regression [40]. It is used to identify and classify patterns or differences between groups in a dataset. PLS-DA is particularly useful when analyzing high-dimensional data where the number of variables is larger than the number of samples. It has applications in various fields, including metabolomics, chemometrics, and bioinformatics. To identify the specific volatile components in rice grains under various storage conditions, we performed a PLS-DA analysis on these components. The score plot showed that the similarity of the six experimental samples was within the 95% confidence interval (Figure A1). In this model, R^2^ (X) = 0.651, R^2^ (Y) = 0.979, and Q^2^ = 0.908, indicating that this model has the advantages of reliability, stability, and good performance. In addition, the rice grain samples were significantly distinguished according to storage conditions in the score plot (Figure A1). A total of 35 characteristic volatile components were detected among three sample groups (Figure A2). Among these volatile compounds, 2,6,10-trimethyl-dodecane (RA: 7.94 ± 1.74 μg/kg, RB: 8.30 ± 1.93 μg/kg, RC: 22.25 ± 1.96 μg/kg), tetradecane (RA: 14.71 ± 1.09 μg/kg, RB: 8.30 ± 1.93 μg/kg, RC: 22.25 ± 1.96 μg/kg), and 2-methyl-tridecane (RA: 9.97 ± 0.86 μg/kg, RB: 12.58 ± 2.74 μg/kg, RC: 11.77 ± 1.35 μg/kg) were detected in the rice grains stored under three different storage conditions, with varying concentrations (Table A1 and Figure A2).

### 3.3. Comparison of Mycotoxins of Rice Grains Stored under Different Conditions

Ochratoxin and aflatoxin are commonly detected mycotoxins that are produced by fungi during the storage of cereals [41]. Moreover, ochratoxin A and aflatoxin B1 are highly toxic and are commonly detected in rice grains [42]. To compare the content of mycotoxins in rice grains under different storage conditions, the content of mycotoxins was determined. Among the three storage conditions, the content of ochratoxin A and aflatoxin B1 under quasi-low temperature, horizontal ventilation, and nitrogen gas atmosphere conditions was 3.67 μg/kg and 3.19 μg/kg, which was the highest (Figure 1H,I). According to international regulations, ochratoxin A and aflatoxin B1 could not be detected higher than 5 μg/kg and 10 μg/kg in samples [43]. The ochratoxin A and aflatoxin B1 of rice grains stored in the three storage conditions did not exceed the national limited safety standards of China and reached the requirement. Previous studies have demonstrated that lower temperatures can effectively inhibit microbial growth and toxin production [44,45]. Additionally, higher temperatures and moisture levels are associated with increased accumulation of aflatoxin B1 and ochratoxin A, with aflatoxin B1 reaching its highest concentration [46]. However, the rice grains stored with the condition of quasi-low temperature had a higher content of ochratoxin A and aflatoxin B1. This may be due to 18 °C also being the optimal condition for Aspergillus to produce toxins [47].

### 3.4. Determination of Microbial Community Structure

A total of 1,177,974 and 1,306,651 partial 16S rRNA and ITS rRNA gene sequences were generated from the 13 rice grain samples, respectively. To compare the richness and diversity of the microbial communities between the rice grains stored under these three storage conditions, Shannon and Chao 1 indices were investigated (Figure 3A,B). The results showed that Shannon (bacteria: 9.16 ± 0.22, fungi: 4.00 ± 0.78) and Chao 1 (bacteria: 2663.16 ± 143.44, fungi: 324.58 ± 263.57) indices of bacteria and fungi communities of the rice grains stored under the condition of normal temperature, vertical ventilation, and nitrogen gas atmosphere were highest among the three groups. It was indicated that in rice grains stored under this condition, elevated microbial activity can contribute to increased external and internal damage, as it becomes more frequent. Principal coordinate analysis (PCoA) was conducted using a Bray–Curtis distance matrix to assess the bacterial and fungal communities’ similarity of rice grains stored under different conditions (Figure 3C,D). The results revealed that the microbial composition in rice grains stored under the condition of normal temperature, horizontal ventilation, and nitrogen gas atmosphere and quasi-low temperature, horizontal ventilation, and nitrogen gas atmosphere exhibited similarities, while the rice grains stored under the condition of normal temperature, vertical ventilation, and nitrogen gas atmosphere displayed distinct differences from the other two storage conditions.

A total of 45 bacteria phyla and 5 fungi phyla were identified in the rice grain samples, among which Proteobacteria (average relative abundance: 52.44%), *Firmicutes* (25.95%), *Acidobacteriota* (5.89%), *Bacteroidota* (4.70%), and *Actinobacteriota* (4.64%) in bacteria and *Ascomycota* (73.06%) and *Basidiomycota* (26.31%) in fungi were dominant, with relative abundance greater than 2% (Figure A4). A total of 854 bacteria genera and 368 fungal genera were detected at the genus level, which further revealed differences in the microbial composition among the three storage conditions. *Pantoea*, *Allorhizobium-Neorhizobium-Parahizobium-Rhizobium*, *Lactobacillus*, and *Pseudocitrobacter* were the most dominant bacterial genera in the rice grain sample. The main genera detected from the microbial community in this study were consistent with previous studies [48]. The relative abundance of *Lactobacillus* (20.47%) and *Alternaria* (26.04%) was highest in the rice grains stored under the condition of normal temperature, horizontal ventilation, and nitrogen gas atmosphere compared with the other two storage conditions (Figure 3E,F). *Lactobacillus* spp. could produce lactic acid, which is often used to ferment food and could also be used as a bacterial inoculant to produce silage and inhibit molds, *Escherichia coli*, etc. [49]. *Alternaria* is frequently isolated from plants [50]. The highest relative abundance of *Pantoea* (14.34%) and *Moesziomyces* (28.72%) was detected in rice grains stored under the condition of normal temperature, vertical ventilation, and nitrogen gas atmosphere storage (Figure 3E,F). *Pantoea* might cause leaf blight disease, which becomes a devastating problem in rice agroecosystems [51]. *Moesziomyces* is often found to parasitize various plants without plant pathogenicity [52]. The relative abundance of *Pantoea* (35.03%) and *Phaeosphaeria* (31.21%) was highest in rice grains stored under quasi-low temperature, horizontal ventilation, and nitrogen gas atmosphere (Figure 3E,F). *Phaeosphaeria* often causes leaf spot disease in tropical and subtropical maize [53]. The high relative abundance of *Pantoea* and *Phaeosphaeria* might cause the degradation of rice grains’ quality. Although several types of microorganisms can be detected, the majority of them are harmless. Some typical filamentous fungi are mainly harmful to rice grains’ quality.

Among the storage conditions under quasi-low temperature, horizontal ventilation, and nitrogen gas atmosphere and normal temperature, vertical ventilation, and nitrogen gas atmosphere, the relative abundance of *Aspergillus* was relatively high, the values of which were 7.81% and 16.02%. Therefore, the highest mycotoxins in rice grains stored under normal temperature, vertical ventilation, and nitrogen gas atmosphere may be due to the highest relative abundance of *Aspergillus*. The quality of rice is affected by some toxigenic fungi such as *Penicillium*, *Aspergillus*, *Fusarium*, and *Alternaria* [54]. Among fungal genera, aflatoxin B1 produced by *Aspergillus flavus* in *Aspergillus* has the highest toxicity. *Aspergillus flavus* can produce aflatoxin B1, which has strong carcinogenicity and teratogenicity and seriously affects the health and safety of mammals [55]. Meanwhile, rice infected by *Aspergillus flavus* is common, which will cause dry matter loss and quality deterioration. This may also be the reason for the low content of starch and protein composition in rice grains stored under normal temperature, vertical ventilation, and nitrogen gas atmosphere. In addition, environmental factors such as temperature, humidity, etc., play determinate roles in the growth of fungi. In the three storage conditions, the relative abundance of *Aspergillus* (1.64%) and *Penicillium* (0.01%) is the lowest in rice grains stored under the condition of normal temperature, horizontal ventilation, and nitrogen gas atmosphere. It was indicated that horizontal ventilation could lower the temperature more than vertical ventilation, thereby inhibiting the growth of harmful fungal generally. The temperature and humidity changes in horizontal ventilation are faster than that in vertical ventilation with a better cooling effect [56]. Therefore, horizontal ventilation should be preferred in the actual storage of rice grains.

To investigate the similarities and differences in microbial communities among rice grains stored under three different conditions, Venn diagrams were constructed (Figure 4A,B). The Venn diagrams showed that the number of shared bacterial ASVs in the three storage conditions was 424 and the common fungal ASVs in the three storage conditions was 141. The rice grains stored under the condition of normal temperature, horizontal ventilation, and nitrogen gas atmosphere contained the most unique ASVs in the bacterial community, which included 1901 ASVs. The rice grains stored under the condition of normal temperature, vertical ventilation, and nitrogen gas atmosphere contained the most unique ASVs in the fungal community, which included 550 ASVs. The common and unique representative genera of rice grains under the three storage conditions were detected (Figure 4A,B). The unique genera of rice grains stored under the condition of normal temperature, vertical ventilation, and nitrogen gas atmosphere were the greatest, with their unique bacterial genera including *Stepomyces*, *Pseudorhodoplanes*, etc., and their unique fungal genera including *Penicillium*, *Stemphylium*, etc. (Figure A3). It was suggested that the storage condition of normal temperature, vertical ventilation, and nitrogen gas atmosphere was more suitable for microbial growth.

### 3.5. Correlations between Physicochemical Indices and the Microbial Community

The correlations between microbial community and the differences in physicochemical indices in the storage process of rice grains were determined (Figure 5). A total of two interaction networks were constructed; one was the correlation between bacterial community and rice grains’ quality, and the other was the correlation between fungal community and rice grains’ quality. Most bacteria genera including *Bacillus*, *Microbacterium*, etc., showed negative correlations with the moisture content, while some ASVs belonging to the genera of *Staphylococcus*, *Sphingomonas*, *Alloprevotella*, etc., showed positive correlations with moisture content. *Staphylococcus aureus* is a pathogenic bacterium and should be given more attention in the process of rice storage. Therefore, moisture content should obtain good control during rice storage. *Sphingomonas* was positively correlated with fatty acid value, which was considered to be an important index for evaluating the suitability of rice grains (Figure 5A). Some bacterial genera including *Bacillus*, *Staphylococcus*, etc., showed significant correlations with albumin, globulin, gliadin, and glutenin (Spearman correlation, *p* < 0.05). According to previous studies, some microorganisms utilize the protein of rice grains to provide nutrition, which destroys the structure of rice grains at the same time [57,58]. Bacteria might have an impact on the quality of rice and are worthy of attention. Some fungi can grow and produce metabolites on rice, causing toxin contamination. From the correlation, the volatile compounds including butyl 1,2-benzenedicarboxylate, 5-methyloctadecane, heptadecane, 4-methylundecane, and hexadecane could be suggested to inhibit *Moesziomyces*. 10-methylesicosane, 2,3,5,8-tetramethyldecane, and *Fusarium* showed negative correlations. *Fusarium* is closely related to *Fusarium* toxins. Butyl 1,2-benzenedicarboxylate and 4-methylundecane might have a bad effect on *Alternaria*. Similarly, butyl 1,2-benzenedicarboxylate and 2-hexyldecanol showed negative correlations with *Microdochium* (Figure 5B). Abnormal activities of microbial communities can lead to an accelerated deterioration in rice quality. Microbial community activities can lead to weight loss, volume reduction, dry matter loss, germination rate reduction, mildew, and even toxin pollution of rice. Therefore, the microbial community is closely related to rice quality. In the future, these volatile components can be developed as green antibacterial ingredients. 

### 3.6. Relationships between Environmental Factors and Microbial Community

Correlations between predominant microbial taxa (top 10 genera with relative abundance of bacterial and fungal communities) and environmental factors (air temperature, air humidity, warehouse temperature, and warehouse humidity) were investigated (Figure 6). Air temperature showed positive correlations with *Pseudarthrobacter* and *Allorhizobium-Neorhizobium-Parahizobium-Rhizobium*, and warehouse temperature positively correlated with *Muribaculaceae*, *Vicinarnibacteraceae*, and *Pseudomonas* (Figure 6A). Warehouse humidity and air temperature showed positive correlations with *Methylobacterium-Methylorubium*, *Lactobacillus*, and *Pseudocitrobacter* (Figure 6A). *Cladosporium* and *Nigrospora* were linked to the varieties of rice grains stored under the condition of normal temperature, horizontal ventilation, and nitrogen gas atmosphere (Figure 6B). *Aspergillus* and *Moesziomyces* were linked to the varieties of rice grains stored under the condition of normal temperature, vertical ventilation, and nitrogen gas atmosphere. In this storage condition, more attention to mycotoxin production should be considered. *Phaeosphaeria* was the contributor to rice grains stored under the condition of quasi-low temperature, horizontal ventilation, and nitrogen gas atmosphere (Figure 6B). Warehouse temperature showed positive correlations with *Nigrospora* and *Phaaeosphaeria* (Figure 6B). Air temperature showed positive correlations with *Candida*, *Moesziomyces*, and *Aspergillus*. *Ustilaginoidea*, *Alternaria*, *Fusarium*, and *Cladosporium* and air humidity showed positive correlations (Figure 6B). Among these environmental factors, air temperature is closely related to some toxigenic filamentous fungi. During rice grain storage, air temperature should be taken into consideration.

The network was constructed to analyze the interaction patterns of co-occurrence or co-exclusion among microbial community individuals in different conditions (Figure 6C,D). *Sphingomonas*, *Vicinamibacteraceae*, and *Muribaculacea* showed positive correlations. *Aspergillus*, *Alternaria*, *Cladosporium*, and *Fusarium* showed positive correlations. Some studies have shown that *Sphingomonas* can produce anticyanobacterial compounds argimicins to form an antagonism with *Escherichia coli*, *Enterobacter* aerogenes, etc., which can also slow down food corruption [59]. *Aspergillus*, *Alternaria*, and *Fusarium* can produce toxins and endanger the quality and safety of rice grains [60]. In the future, more attention should be paid to the changes in the abundance of these genera.

## 4. Conclusions

The application of different storage conditions of rice grains caused differences in rice grains’ quality and microbial communities. The storage environmental indicators, including air temperature, air humidity, granary temperature, and granary humidity, significantly influence the composition of microbial communities, including *Lactobacillus*, *Pseudocitrobacter*, *Aspergillus*, *Fusarium*, etc., in rice grains during the storage period, and the microorganisms present in rice grains, such as albumin, globulin, etc., have a notable impact on their quality. In the subsequent joint analysis, it was also found that butyl 1,2-benzenedicarboxylate was negatively correlated with *Moesziomyces* and *Alternaria*. 10-methylesicosane, 2,3,5,8-tetramethyldecane, and *Fusarium* showed negative correlations. In the future, we can further verify the effect of these volatile components on fungi and develop green biological bacteriostatic agents. This study provides a microbiological basis for understanding how different storage environments can lead to variations in rice grains’ quality, and it offers guidance for achieving better storage of rice grains.

## Figures and Tables

**Figure 1 foods-13-00266-f001:**
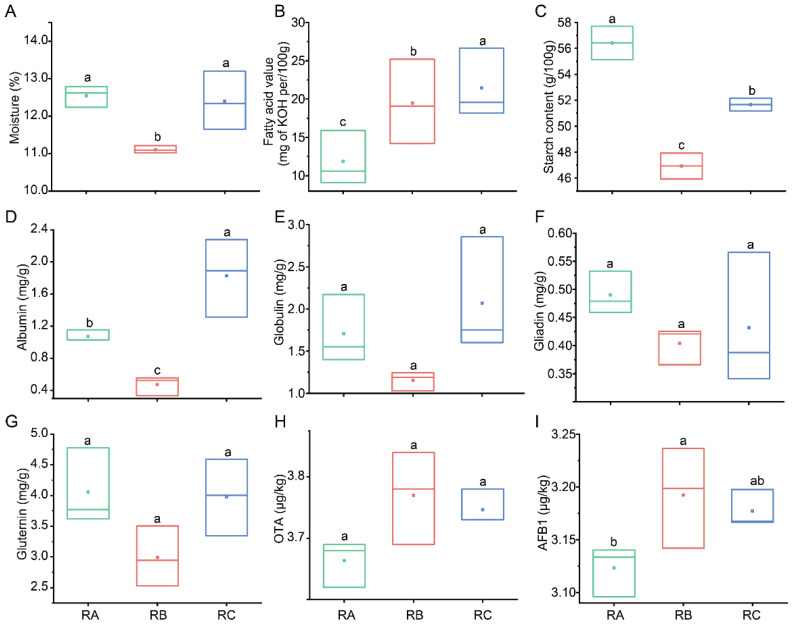
Changes in the chemical substance and mycotoxin contents of rice grains stored under three different storage conditions. (**A**) Moisture. (**B**) Fatty acid value. (**C**) Starch content. (**D**) Albumin. (**E**) Globulin. (**F**) Gliadin. (**G**) Glutenin. (**H**) OTA. (**I**) AFB1. RA: room temperature, horizontal ventilation, nitrogen gas atmosphere; RB: room temperature, vertical ventilation, nitrogen gas atmosphere; RC: quasi-low temperature, horizontal ventilation, nitrogen gas atmosphere. Different letters indicate significant difference (*p* < 0.05) among contents of rice grain in three storage environments.

**Figure 2 foods-13-00266-f002:**
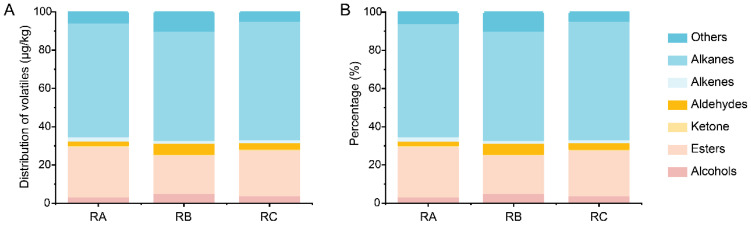
The composition of volatile components in rice grains stored under three different conditions. Content (**A**) and proportion (**B**) of different volatile components. “Others” represents aromatic and heterocyclic compounds. RA: room temperature, horizontal ventilation, nitrogen gas atmosphere; RB: room temperature, vertical ventilation, nitrogen gas atmosphere; RC: quasi-low temperature, horizontal ventilation, nitrogen gas atmosphere.

**Figure 3 foods-13-00266-f003:**
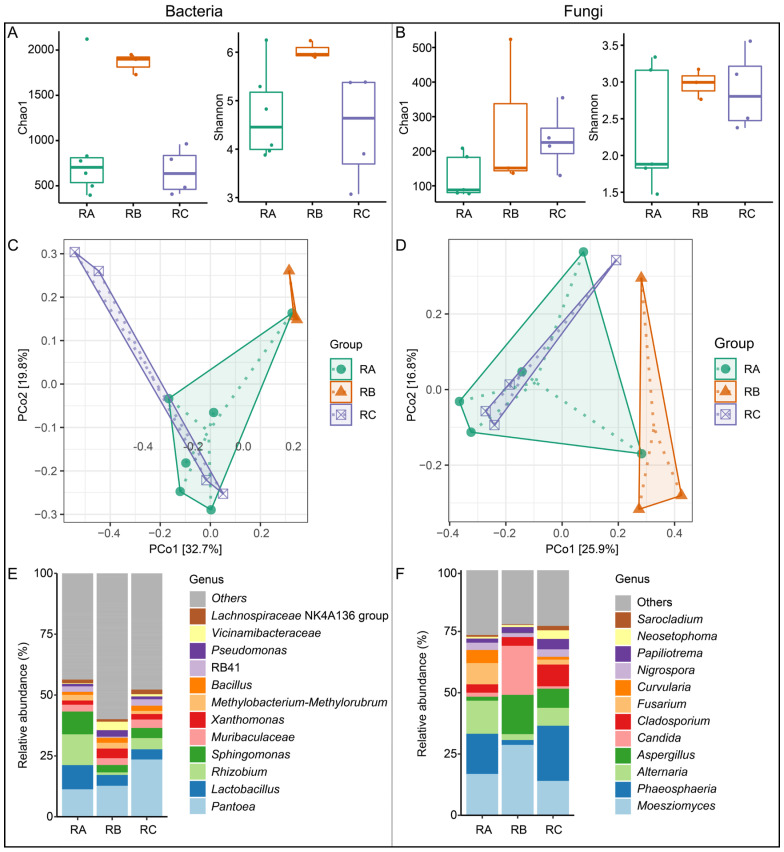
Bacterial and fungal community structures of rice grain samples in different storage environments. The Chao 1 index and Shannon index of bacteria (**A**) and fungi (**B**). Principal coordinates analysis (PCoA) of the bacterial (**C**) and fungal community (**D**) based on Bray–Curtis distance. Relative abundance of bacteria (**E**) and fungi (**F**). RA: room temperature, horizontal ventilation, nitrogen gas atmosphere; RB: room temperature, vertical ventilation, nitrogen gas atmosphere; RC: quasi-low temperature, horizontal ventilation, nitrogen gas atmosphere.

**Figure 4 foods-13-00266-f004:**
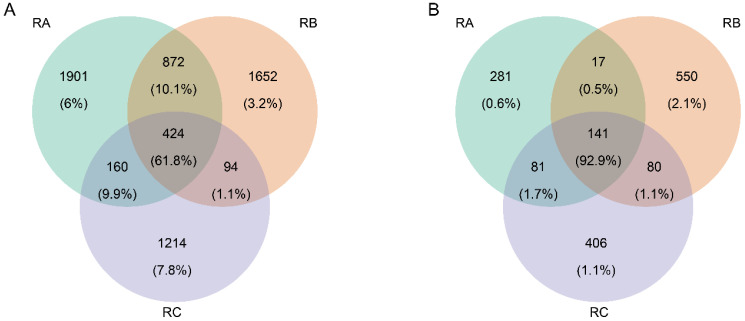
Shared and unique microbial community in rice grains stored under three storage environments. Venn diagrams show the number of shared and unique bacterial (**A**) and fungal ASVs (**B**). RA: room temperature, horizontal ventilation, nitrogen gas atmosphere; RB: room temperature, vertical ventilation, nitrogen gas atmosphere; RC: quasi-low temperature, horizontal ventilation, nitrogen gas atmosphere.

**Figure 5 foods-13-00266-f005:**
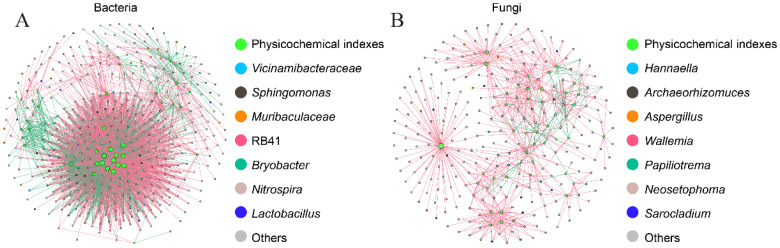
Correlations between bacterial (**A**) and fungal community (**B**) and rice grains’ quality, including moisture content, starch content, fatty acid value, protein composition, volatile compounds (VIP > 1), and mycotoxin content (*p* < 0.05).

**Figure 6 foods-13-00266-f006:**
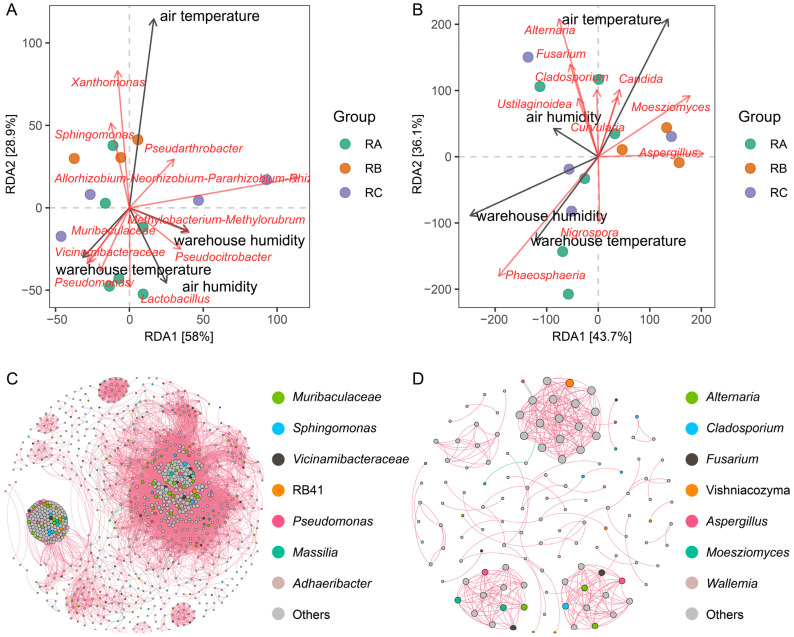
Relationships between predominant bacterial (**A**) and fungal genera (**B**) and environmental factors (air temperature, air humidity, warehouse temperature, and warehouse humidity). Co-occurrence networks based on Spearman correlation coefficients of bacterial (**C**) and fungal genera (**D**) in rice grains stored under different conditions. RA: room temperature, horizontal ventilation, nitrogen gas atmosphere; RB: room temperature, vertical ventilation, nitrogen gas atmosphere; RC: quasi-low temperature, horizontal ventilation, nitrogen gas atmosphere.

**Table 1 foods-13-00266-t001:** The storage conditions of rice grains.

Storage Environment	Samples	Granary Temperature (℃)	Air Temperature (℃)	Granary Humidity (%)	Air Humidity (%)	Grain Depot
RA	RA.1	13.6	14	50.9	82	Shaoxing
RA	RA.2	13.5	8	59.75	82	Shaoxing
RA	RA.3	12.4	8	68.9	82	Shaoxing
RA	RA.4	12.8	8	61.2	82	Ningbo
RA	RA.5	12.8	8	61.2	82	Ningbo
RA	RA.6	15.4	8	68.9	82	Ningbo
RB	RB.1	12	8	70.7	82	Quzhou
RB	RB.2	11.5	8	71	82	Quzhou
RB	RB.3	11.4	8	66.8	82	Quzhou
RC	RC.1	13.7	8	64.9	82	Hangzhou
RC	RC.2	13.5	8	63.5	82	Hangzhou
RC	RC.3	13.5	15	63.5	82	Hangzhou
RC	RC.4	15	8	68.4	82	Hangzhou

Note: “RA” represents that paddy was stored at room temperature, horizontal ventilation, and nitrogen gas atmosphere; “RB” represents that paddy was stored at room temperature, vertical ventilation, and nitrogen gas atmosphere; “RC” represents that paddy was stored at quasi-low temperature, horizontal ventilation, and nitrogen gas atmosphere.

**Table 2 foods-13-00266-t002:** The content of volatile compounds of rice grains stored under different environments (μg/kg).

N0	Compound Name	Storage Conditions (Mean ± SD)		
		RA	RB	RC
Alcohols				
1	2-Ethyl-1-hexanol	15.78 ± 1.41 a	22.04 ± 4.61 a	15.52 ± 0.62 a
2	2-Hexyl-1-decanol	ND	5.59 ± 0.52 a	1.81 ± 0.13 b
3	2-Butyl-1-octanol	ND	4.85 ± 2.02 a	ND
4	2-Isopropyl-5-Methyl-1-heptanol	ND	ND	5.54 ± 1.68 a
Acid esters				
1	1,2-Benzenedicarboxylic acid, bis(2-methylpropyl) ester	7.20 ± 4.76 a	4.03 ± 1.19 a	3.61 ± 0.46 a
2	Ethyl dodecanoate	6.6 ± 0.06 a	6.41 ± 1.71 a	6.75 ± 1.56 a
3	Ethyl heptanoate	18.40 ± 3.15 a	16.87 ± 6.78 a	16.02 ± 4.06 a
4	Ethyl hexadecanoate	5.02 ± 2.18 a	23.40 ± 7.52 a	18.20 ± 12.49 a
5	Ethyl nonanoate	48.72 ± 2.50 a	30.17 ± 5.67 a	41.68 ± 13.50 a
6	Ethyl octanoate	22.65 ± 7.14 a	16.61 ± 5.63 a	21.05 ± 5.04 a
7	Ethyl myristate	6.78 ± 0.37 a	5.67 ± 2.56 a	14.74 ± 12.67 a
8	1,2-Benzenedicarboxylic acid, butyl ester	ND	3.81 ± 0.81 a	3.06 ± 0.25 a
9	Sulfurous acid, dodecyl pentyl ester	ND	9.3 ± 0.21 a	ND
10	Ethyl 9-hexadecenoate	ND	ND	1.58 ± 0.47 a
11	Sulfurous acid, 2-pentyl undecyl ester	10.98 ± 0.71 a	ND	ND
Ketone				
1	6,10,14-Trimethyl-2-pentadecanone	8.51 ± 2.72 a	6.17 ± 1.33 a	8.67 ± 2.81 a
Aldehydes				
1	Nonanal	17.55 ± 5.58 a	36.15 ± 10.09 a	20.42 ± 7.65 a
Alkenes				
1	1-Dodecene	3.39 ± 0.24 a	ND	ND
2	4,6,8-Trimethyl- 1-nonene -1-Nonene,	3.57 ± 0.59 a	3.99 ± 1.02 a	ND
3	1-Tetradecene	4.28 ± 0.50 a	4.78 ± 0.35 a	4.45 ± 0.05 a
Hydrocarbons				
1	2,3,5,8-Tetramethyldecane-	6.80 ± 7.34 a	ND	ND
2	Dodecane	11.45 ± 1.19 a	15.04 ± 3.12 a	13.02 ± 1.47 a
3	2,6,10-Trimethyldodecane	7.94 ± 1.74 b	8.3 ± 1.93 b	22.25 ± 1.96 a
4	2,6,11-Trimethyldodecane	51.61 ± 6.01 a	54.22 ± 11.98 a	45.62 ± 18.80 a
5	4-Cyclohexyldodecane Dodecane	7.26 ± 1.00 a	7.32 ± 0.61 a	7.39 ± 0.51 a
6	5-Methyldodecane	6.18 ± 0.25 a	5.80 ± 0.03 a	ND
7	Eicosane	29.75 ± 3.49 a	51.58 ± 19.06 a	35.92 ± 18.65 a
8	10-Methyl- eicosane Eicosane	5.32 ± 0.10 a	ND	ND
9	Heptadecane	ND	11.77 ± 1.21 a	15.09 ± 7.07 a
10	2,6,10,15-Tetramethylheptadecane	5.37 ± 0.8 a	ND	ND
11	2,6,10,14-Tetramethylhexadecane	5.13 ± 0.98 a	ND	ND
12	Nonadecane	22.81 ± 12.74 a	22.35 ± 6.78 a	20.32 ± 7.82 a
13	Pentadecane	15.38 ± 13.30 a	17.14 ± 3.74 a	21.35 ± 6.26 a
14	2-Methylpentadecane	2.64 ± 0.10 a	2.51 ± 0.35 a	2.82 ± 0.22 a
15	3-Methylpentadecane	7.82 ± 0.07 a	8.08 ± 0.28 a	7.90 ± 0.75 a
16	Tetradecane	14.71 ± 1.09 a	6.84 ± 0.54 b	7.42 ± 1.26 b
17	Tridecane	5.78 ± 1.13 a	ND	ND
18	2-Methyltridecane	9.97 ± 0.86 a	12.58 ± 2.74 a	11.77 ± 1.35 a
19	3-Methyltridecane	10.12 ± 1.15 a	12.58 ± 2.74 a	11.77 ± 1.35 a
20	4-Methyltridecane	1.96 ± 0.18 a	2.60 ± 0.04 a	2.54 ± 0.27 a
21	2,4-Dimethylundecane	17.76 ± 0.92 a	16.91 ± 1.87 a	14.85 ± 5.03 a
22	2,6-Dimethylundecane	8.58 ± 0.89 a	11.63 ± 4.03 a	10.60 ± 3.33 a
23	3-Methyleneundecane	1.85 ± 0.40 b	6.11 ± 2.24 a	2.13 ± 0.44 b
24	4,8-Dimethylundecane	5.12 ± 1.52 a	7.14 ± 3.80 a	6.75 ± 1.15 a
25	3,5-Dimethyldodecane	ND	2.56 ± 0.37 a	ND
26	Cyclotridecane	ND	3.80 ± 0.03 a	ND
27	4-Ethyldecane	ND	2.92 ± 0.45 a	ND
28	Heneicosane	ND	5.62 ± 0.19 a	ND
29	2-Methylheptadecane	ND	1.67 ± 0.95 a	ND
30	Hexadecane	ND	11.77 ± 1.21 a	15.09 ± 7.07 a
31	4-Methylhexadecane	ND	2.03 ± 0.31 a	ND
32	4,5-Dimethylnonane	ND	10.69 ± 4.66 a	8.61 ± 3.84 a
33	Octadecane	ND	4.72 ± 0.72 a	ND
34	5-Methyloctadecane	ND	9.17 ± 3.04 a	10.90 ± 3.42 a
35	4-Methylundecane	ND	3.02 ± 0.92 a	2.47 ± 0.27 a
36	5,7-Dimethylundecane	ND	5.32 ± 4.83 a	ND
37	2-Methyldodecane	ND	ND	1.16 ± 1.01 a
38	4,6-Dimethyldodecane	ND	ND	7.06 ± 4.11 a
39	5-Methyldodecane	ND	ND	5.61 ± 0.03 a
40	11-(1-Ethylpropyl)-heneicosane	ND	ND	2.79 ± 0.16 a
41	2,6,11,15-Tetramethylhexadecane Hexadecane	ND	ND	4.01 ± 2.4 a
42	2,5-Dimethyltridecane	ND	ND	3.5 ± 1.53 a
43	4,8-Dimethyltridecane	ND	ND	5.32 ± 0.1 a
44	3-Methyleneundecane	ND	ND	3.96 ± 1.47 a
Others				
1	1,3-Dichlorobenzene	19.07 ± 17.9 b	50.8 ± 3.13 a	ND
2	2,4-Di-tert-butylphenol	13.53 ± 1.20 a	14.22 ± 0.96 a	ND
3	2-Pentylfuran	ND	10.57 ± 0.44 a	ND

Note: RA: room temperature, horizontal ventilation, nitrogen gas atmosphere; RB: room temperature, vertical ventilation, nitrogen gas atmosphere; RC: quasi-low temperature, horizontal ventilation, nitrogen gas atmosphere. Different letters indicate significant difference (*p* < 0.05) among contents of rice grain in three storage environments. “ND” means the substance was not detected.

## Data Availability

The data presented in this study are available on request from the corresponding author. The data are not publicly available due to a parallel research project that is currently unpublished.

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
