# Peer review of "Deciphering the Microbiological Mechanisms Underlying the Impact of Different Storage Conditions on Rice Grain Quality"

_foods, 2024, doi:10.3390/foods13020266_

Round 1
Reviewer 1 Report
Comments and Suggestions for Authors
I would like to give my congratulations to the authors for their research results.
Also, I would recommend them to check the comments in the manuscript attached.

The language needs further improvement.
Reviewer 2 Report
Comments and Suggestions for Authors
Review of the manuscript titled “Deciphering the Microbiological Mechanisms Underlying the 2 Impact of Different Storage Conditions on Rice Grain Quality”.
The study is devoted to an important subject of rice quality after storage under different conditions. The study employs very sophisticated methods to evaluate the grain quality and its microbiome. The statistical procedures and presentations of the results are outstanding. The study also covers all aspects of grain quality. The paper contributes substantially to the scientific subject and deserves publication. The paper is also very well written and gives comprehensive picture of the subject.
However, there is a major deficiency in the paper which needs to be solved to have it published in the current format. The authors attribute all the changes in the rice grain to different storage procedures without the analysis of the grain prior to storage. All the grain attributes analyzed in the study are affected by the cultivars, production technologies, weather at harvest, soil conditions and other factors. The authors try to explain the differences between samples to three storage conditions which are, in fact, quite similar. What is the logical explanation of the differences between RA and RB treatments which is horizontal versus vertical ventilation. If ventilation is efficient – maybe it does not matter which way it goes. The authors did not evaluate the grain prior to storage and did not establish the baseline to track real differences due to storage conditions. Ideally, it should be several identical grain samples exposed to different storage conditions. Then the real differences due to storage can be documented. Now, there are two ways to solve this problem: a) authors can argue and demonstrate that the storage conditions are more important than all other factors which contribute to grain quality and its microbiome; perhaps they will be successful; b) focus on describing very valuable data of relationship between microbial communities and rice quality in the context of samples origin, may be growing conditions, varieties and also storage conditions as one of the factors. There is so much valuable data which authors can use without any need to relate them to storage conditions.
The authors can also check if the storage conditions significantly affect the quality parameters. They need to make ANOVA for three treatments may be using samples as replications. If the treatments significantly affect the target traits – they can argue that the differences were indeed due to storage. However, here the geographical origin of the samples needs to be distributed among the treatments.
There are few other comments which can improve the paper.
1. Introduction. Current common practices of rice storage in China need to be presented as a context of the study. Also the share of poor quality rice due to poor storage would be useful to mention.
2. Section 2.1. A more detailed explanation is needed: the period of exposure to the storage conditions; what is the difference between the granary (storage place or actually grain) and air temperature and the same for humidity; how horizontal and vertical ventilation were provided including the volume or speed; the size of the chamber where the samples were exposed to different conditions; what is “nitrogen gas atmosphere”. It is not clear if 13 samples collected from four granaries were simply analyzed by the authors or they were collected and then exposed to different treatments.
3. “Quasi low temperature” – what does it mean? According to the Table 1, RC variant has the same temperature as RA.
4. Table 1 – the geographical origin of the samples needs to be added.
5. Section 2.3.1. What is OTA and AFB?
Reviewer 3 Report
Comments and Suggestions for Authors
The aim of this study was to investigate how various storage conditions, particularly those utilizing nitrogen gas atmosphere (RA, RB, RC), influence the quality of rice grains. Specifically, the study aimed to understand the microbial community structures and metabolic functions affecting rice quality under these conditions. By monitoring factors such as fungal toxins, volatile compounds, and microbial abundance, the research aimed to identify the impact of storage conditions on rice quality. At the first glance the aim looks good but there are some points that should be considered before publication:
Please don’t use the same words in the keywords which you have already used in the title.
Table 1: How could you define these conditions? It is not clear. Please clarify.
Fig.1 and Table 1: For some parameters look with high standard deviations. At first could you please provide the reason behind of it in the text. Do you think two replicate is enough to show the high variation?
Fig.2 c and fig. 4 c and d can be seen. Please improve their quality. You can also think that put them in the appendix.
Conclusion: Please provide core points that you obtained in the work. It is so short now.
Generally, figures are not well and so confusing. Please think and find a way of organizing them.
Comments on the Quality of English LanguageModerate editing of English language required.
Round 2
Reviewer 2 Report
Comments and Suggestions for Authors
The authors substantially improved the manuscript and it is now ready for publication.
Reviewer 3 Report
Comments and Suggestions for Authors
From my side, it can be publishable.
Comments on the Quality of English LanguageMinor editing of English language required.